# Estimating Minimal Clinically Important Differences for Knee Range of Motion after Stroke

**DOI:** 10.3390/jcm9103305

**Published:** 2020-10-15

**Authors:** Agnieszka Guzik, Mariusz Drużbicki, Andżelina Wolan-Nieroda, Andrea Turolla, Pawel Kiper

**Affiliations:** 1Department of Physiotherapy, Institute of Health Sciences, Medical College, University of Rzeszów, 35-959 Rzeszów, Poland; mdruzb@ur.edu.pl (M.D.); wolan.a@gmail.com (A.W.-N.); 2Laboratory of Kinematics and Robotics IRCCS San Camillo Hospital, 30126 Venice, Italy; andrea.turolla@ospedalesancamillo.net; 3Azienda ULSS 3 Serenissima Physical Medicine and Rehabilitation Unit, 30126 Venice, Italy; pawelkiper@hotmail.com

**Keywords:** stroke, minimal clinically important difference, gait, hemiplegia

## Abstract

The importance of knee sagittal kinematic parameters, as a predictor of walking performance in post-stroke gait has been emphasised by numerous researchers. However, no studies so far were designed to determine the minimal clinically important differences (MCID), i.e., the smallest difference in the relevant score for the kinematic gait parameters, which are perceived as beneficial for patients with stroke. Studies focusing on clinically important difference are useful because they can identify the clinical relevance of changes in the scores. The purpose of the study was to estimate the MCID for knee range of motion (ROM) in the sagittal plane for the affected and unaffected side at a chronic stage post-stroke. Fifty individuals were identified in a database of a rehabilitation clinic. We estimated MCID values using: an anchor-based method, distribution-based method, linear regression analysis and specification of the receiver operating characteristic (ROC) curve. In the anchor-based study, the mean change in knee flexion/extension ROM for the affected/unaffected side in the MCID group amounted to 8.48°/6.81° (the first MCID estimate). In the distribution-based study, the standard error of measurement for the no-change group was 1.86°/5.63° (the second MCID estimate). Method 3 analyses showed 7.71°/4.66° change in the ROM corresponding to 1.85-point change in the Barthel Index. The best cut-off point, determined with ROC curve, was the value corresponding to 3.9°/3.8° of change in the knee sagittal ROM for the affected/unaffected side (the fourth MCID estimate). We have determined that, in chronic stroke, MCID estimates of knee sagittal ROM for the affected side amount to 8.48° and for the unaffected side to 6.81°. These findings will assist clinicians and researchers in interpreting the significance of changes observed in kinematic sagittal plane parameters of the knee. The data are part of the following clinical trial: Australian New Zealand Clinical Trials Registry: ACTRN12617000436370

## 1. Introduction

Measurement of progress in rehabilitation is of utmost importance. For post-stroke rehabilitation, there are numerous standardised clinical measures predominantly assessing temporal and spatial gait parameters, e.g., 10-m walk test [1,2], 2-, 6-, and 12-min walk tests [2,3], up and go test [4], cadence [5] or step length symmetry [6]. Furthermore, the concept of minimal clinically important difference (MCID) is currently at the centre of a newer approach in research on clinical measures post-stroke [7]. MCID is defined as “the smallest difference in score in the domain of interest which patients perceive as beneficial and which would mandate (...) a change in the patient’s management” [8]. The values of MCID have already been determined for the above measures [9,10,11]. Conversely, no studies have aimed to determine the MCID for kinematic gait parameters post-stroke, even though kinematic analysis of gait provides objective evidence and may successfully be used to evaluate effects of gait re-education [12].

The knee function is critical to walking because it enables the lower leg to move relative to the thigh, and provides support for the weight of the body. Furthermore, the motion in the knee also adjusts the length of the lower limb during the swing phase [13,14]. Impaired knee kinematics is a common consequence of stroke, as a result of which normal mobility becomes impossible, e.g., most patients with hemiparesis after stroke present with knee hyperextension during the subphases of loading response and at mid stance [15].

Evaluation of gait kinematics post-stroke is an important part of assessment. The measure is essential for planning rehabilitation, and for monitoring the effectiveness of therapies [16,17]. Clinical gait analysis generally applies 3-dimensional (3D) kinematic measures of gait, which are also used as key outcome measures in clinical practice and in gait research [18].

Many researchers have pointed out that the knee function is of essential importance in describing and monitoring changes in post-stroke gait patterns [19,20,21]. Taking into account the sagittal knee parameters, i.e., knee flexion and knee extension, Beyaert et al. identified three main types of patterns in individuals with chronic stroke. Significant and prolonged hyperextension of the affected knee during stance, followed by low peak knee flexion in swing phase, can frequently be observed in very slow gait. Significant and prolonged flexion of the affected knee during stance, followed by irregular decrease in knee flexion in swing phase may commonly be observed in very slow or slow gait. Mild to moderate flexion or hyperextension of the affected knee during part of stance, followed by irregular mild reduction in knee flexion in the swing phase can usually be observed in moderate or faster gait [22].

It has been established that the highest reliability indices occur in the hip and knee in the sagittal plane [16,20], with the lowest errors in pelvic rotation and obliquity as well as hip abduction [16]. However, MCID has not yet been calculated for any of the above major predictors of walking performance post-stroke. Due to this we have decided to start the related research by identifying the MCID for knee flexion/extension range of motion (ROM) for the affected and the unaffected side. We chose to focus on this parameter first because it permanently affects post-stroke gait and numerous researchers have emphasised the importance of knee sagittal kinematic parameters, as a predictor of walking performance post-stroke [18,20,23,24,25,26]. Moreover, the sagittal knee pattern in individuals with hemiparesis after stroke strongly correlates with the sagittal ankle or foot patterns and to a lesser degree to the sagittal hip pattern [20,22,25,27,28]. This conclusion provided a motivation for the present study.

The study aimed to estimate MCID values for knee ROM in the sagittal plane for the affected and the unaffected side in the late period post-stroke. We estimated MCID values using: a patient anchor-based method, distribution-based method, linear regression analysis and specification of the receiver operating characteristic (ROC) curve.

## 2. Materials and Methods

### 2.1. Participants

The study involved 57 patients in a chronic phase of recovery post stroke recruited among the 200 patients with stroke receiving treatment at a rehabilitation clinic (117 patients were not eligible to participate and 26 patients were not interested after receiving information about the study protocol). Ultimately the analyses took into account 50 individuals (7 patients were not assessed at the follow-up because their rehabilitation program was completed earlier than four weeks after the initial assessment). The eligibility criteria were defined as follows: age 30–75 years; single ischaemic stroke; a minimum of 6 months from the stroke incident; unilateral paresis; Brunnström recovery stage 3–4; Functional Ambulation Category level 3 or higher, with an ability to get up from chair without help, and walk a distance of at least 10 m without assistance (walking aid permitted, walking speed > 0.4 m/s). Patients excluded from the study had more than one stroke incident, presented unstable medical condition, orthopaedic disorders of the lower limbs, pain and inflammation in the musculoskeletal system significantly affecting gait and requiring anti-inflammatory drugs, cognitive impairment affecting their ability to understand the instructions, and perform the tasks. The study was approved by the local Bioethics Commission of the Medical Faculty University of Rzeszow (5/2/2017, date: 9 February 2017). All procedures were carried out in compliance with the Declaration of Helsinki. Written informed consent was obtained from all the subjects. No adverse events were observed during the study. The flow of the subjects through the study is presented in Figure 1. The baseline subject characteristics are shown in Table 1. The knee kinematic characteristics of the study participants are shown in Table 2. Representative graphs for knee flexion/extension ROM for the affected and unaffected sides in two subjects with right- and left-hemiparesis are shown in Figure 2.

### 2.2. Measures

Kinematic knee data were collected with a six-camera motion capture system (BTS SMART-DX 700, 250 Hz) with software, in SMART Capture, Tracker and Analyzer and two force-plates. Passive markers were placed on the subjects’ skin, following the internal protocol of the system Davis Marker Placement, on the sacrum, pelvis (the anterior and posterior iliac spine), femur (lateral epicondyle, great trochanter and in the lower one-third of the shank), fibula (lateral malleolus, lateral end of the condyle in the lower one-third of the shank), as well as foot (metatarsal head and heel) [29]. Each 3D assessment was preceded by the system calibration. The recording of each patient covered a walking distance of 10 m, repeated at least six times. More trials were necessary if the patient lost balance or excessively hesitated during the basic trials. The tests were conducted without shoes. The subjects were asked to walk the distance at their natural pace. A 3D skeletal model was created for each subject. The model and the joint centres were scaled, taking into account the subject’s height and weight. The generic model also was scaled. After the tests were performed, the data were collected and processed with software from the BTS Smart system (Smart Tracker and Smart Analyzer). The analyses took into account the complete range of knee flexion and extension in a gait cycle for the affected and the unaffected side. The gait cycle for each leg was defined to comprise all the phases starting with heel strike and ending with the next contact of the same foot with the ground. One stance phase and one swing phase were recorded during a single gait cycle performed by each leg. The analyses took into account a minimum of six gait cycles performed by each subject. Based on that, mean values of biomechanical gait parameters were calculated for the complete range of knee flexion and extension for the affected and unaffected side.

Barthel Index (BI) is a tool designed for assessing activities of daily living in patients with various impairments. It takes into account 10 aspects of daily living, e.g., grooming, toilet use, and ambulation. BI is the basic tool applied in clinical practice in assessing functional performance post-stroke [30,31]. The MCID for BI post-stroke was estimated to be 1.85 points [32]. In this study, BI was determined twice, at baseline and at follow-up.

### 2.3. Data Analysis

Statistica 13.1 (StatSoft, Cracow, Poland) was used to compute all the statistics. Descriptive statistics (mean, standard deviation) were calculated for the participants’ knee kinematic characteristics. A significance level was assumed for *p* < 0.05. The statistical comparisons applied mean difference and a 95% CI.

The MCID for the knee ROM, for the affected/unaffected side, was determined using four methods, and finally the highest result was selected.

The anchor-based method made it possible to identify the first estimate for the MCID, i.e., the mean change in the knee sagittal ROM for the affected/unaffected side in the “positive change group” (MCID group). Anchor-based methods take into account change in scores related to a clearly defined clinical observation. External criteria applied include perception of the change by the patient or clinician. Besides a simple estimate of change, the construct of “important change” implies that a change has occurred and is perceived as significant by the patient, physician, or researcher [33,34,35,36].

The distribution-based method was used to determine the second estimate for the MCID, i.e., the standard error of measurement (SEM) was computed as the square root of the variance of a change in the knee sagittal ROM in the relevant subgroup. Distribution-based methods take into account the statistical characteristics of the scores obtained, i.e., their significance, or sample variation, or measurement precision. Representing the latter type, the method of the standard error of measurement (SEM) is most promising for MCID-related research for three reasons. It takes into account both the amount of error specific to the instrument and the amount of random variation to be expected in repeated administrations. It is not greatly affected by the sample size or change variability. Finally, it is sample-independent [32,37].

Linear regression analysis was applied to identify the third estimate for the MCID, i.e., the relationship between the change in the knee sagittal ROM (dependent variable) and the change in BI (independent variable). Linear regression is applied in various analyses. The biggest advantage of linear regression models is linearity: it makes the estimation procedure simple and, most importantly, these linear equations are easy to interpret on a modular level [38,39].

ROC curve, applied to determine the fourth estimate of the MCID in the knee ROM for the affected/unaffected side, made it possible to identify the optimal cut-off point for the change in the knee sagittal ROM, producing the optimum relation of sensitivity and specificity. ROC curves are often applied to visualise the connection/trade-off between sensitivity and specificity for every possible cut-off for a test or a combination of tests. Sensitivity is described as a probability that if a rule says an event will occur, it indeed will occur. Specificity on the other hand is a probability that if a rule says an event will not happen, it indeed will not happen. When we have calculated sensitivity and specificity, we may draw an ROC curve for every possible cut-off. Generally, it is impossible to have high sensitivity and high specificity at the same time, but we strive for perfection. Hence, we want our ROC curve to get as close as possible to the left upper corner, to identify a point in that area. In terms of sensitivity and specificity, this point most accurately corresponds to the change identified as MCID [40,41].

In order to assess the age-related effects, comparative analyses were performed taking into account groups distinguished based on the category of age (dichotomous age stratification into two groups of subjects up to 50 and over 50 years of age). For this purpose, descriptive statistics were compared and differences between the average values were assessed (Mann–Whitney test) by examining the relations between age and the knee flexion/extension ROM identified before the start of the rehabilitation program and showing effects of rehabilitation.

### 2.4. Procedures

All patients participated in a four-week rehabilitation programme, comprising individual and group exercises, ergotherapy, and psychotherapy. The therapy addressed the basic motor functions, change of position, standing up, walking, and balance. On average, each patient exercised for two hours daily.

For the needs of the anchor-based method, knee ROM was assessed at baseline (upon the patients’ admission to the rehabilitation department) and at a follow-up (four weeks after the initial assessment). During the latter assessment, the subjects were asked whether they noticed any actual change in their knee ROM, compared to their condition before the therapy. Ultimately, three groups were distinguished based on the subjects’ perceptions of the improvement achieved in the knee ROM. Those reporting “no change” were assigned to the “no-change group”. Those reporting improvement were allocated to the “MCID group” (or “positive change group”). Those who claimed their condition was “worse,” were classified into the “negative change group”.

The distribution-based method was applied in analyses related to the no-change group, which means that SEM was calculated for the patients reporting no change in their knee sagittal ROM between baseline and the follow-up.

A linear regression analysis was used to assess a change in the knee sagittal ROM by comparison to a clinically important change in the BI.

The ROC curve method was used in analyses taking into account the “no-change group” and the “positive change group”, to identify the cut-off point for the change in the knee sagittal ROM most effectively separating these two groups with the ROC curve.

## 3. Results

Taking into account the anchor-based analyses, only two groups (MCID/positive change and no-change) were distinguished based on the patients’ perceptions of the knee sagittal ROM, because no subjects reported deterioration in the knee sagittal ROM at the follow-up. The no-change group comprised 16 patients, while the MCID/positive change group consisted of 34 subjects. The mean values of knee sagittal ROM for the affected/unaffected side in the MCID group and no-change group are shown in Table 3.

For the needs of the anchor-based estimate, only the changes identified in the MCID group were analysed. The mean change in knee flexion/extension ROM for the affected side was 8.48°, which constituted the first estimate of the MCID of the knee flexion/extension ROM for the affected side (Table 4). The mean change in knee flexion/extension ROM for the unaffected side was 6.81°, which constituted the first estimate of the MCID of knee flexion/extension ROM for the unaffected side (Table 5).

Distribution-based analyses took into account the no-change group only. The SEM of 1.86° was adopted as the second estimate of the MCID of the knee sagittal ROM for the affected side (Table 4). The SEM of 5.63° was the second estimate of the MCID of the knee sagittal ROM for the unaffected side (Table 5).

Linear regression analyses showed the slope of the regression line amounting to 4.168° which means than a one-point change in the BI is associated with a change of 4.168° in the knee sagittal ROM for the affected side (Figure 3). This translates to a change of 7.71° in the range of motion corresponding to a 1.85-point change in BI (the third MCID estimate of the knee sagittal ROM for the affected side)—Table 4. The slope of the regression line was 2.518° which means than one-point change in BI is associated with a change of 2.518° in the knee sagittal ROM for the unaffected side (Figure 4). This translates to a change of 4.66° in the range of motion corresponding to a 1.85-point change in BI (the third MCID estimate of the knee sagittal ROM for the unaffected side)—Table 5.

The analyses based on ROC curve showed that the best cut-off points were the values representing 3.9° and 3.8° change in the knee sagittal ROM for the affected and unaffected side, respectively (Figure 5 and Figure 6)—the fourth estimate of the MCID for the knee sagittal ROM for the affected/unaffected side (Table 4 and Table 5).

Given the fact that the subjects participating in the study represented a wide age range, they were divided into two groups—up to 50 (7 subjects –14%) and over 50 years of age (43 subjects—86%). A comparison based on the dichotomous distinction of the age groups showed there were no relationships between age and the knee flexion/extension ROM in the patients prior to the rehabilitation (Table 6). Likewise, a comparative analysis of the results achieved by these two age groups showed no statistically significant differences in the effects of the rehabilitation (Table 6).

The highest of the four MCID estimates of the knee sagittal ROM for the affected side amounted to 8.48° and for the unaffected side it was 6.81°, in patients with chronic stroke, which is defined as the MCID for these respective parameters.

## 4. Discussion

This study aimed to identify MCID for knee sagittal ROM separately for the affected and for the unaffected side. The concept of MCID has been investigated in the context of stroke [7], however, to the best of our knowledge, the current study was the first to determine the MCID for the knee sagittal ROM in patients with stroke. It was shown that, in chronic stroke, MCID estimates of the knee sagittal ROM for the affected and unaffected side amounted to 8.48° and 6.81°, respectively. Notably, the estimated MCIDs were slightly lower for the unaffected side. Indeed, it seems logical that MCID in knee sagittal ROM will be smaller for the unaffected than for the affected side. Unfortunately, we cannot compare the current findings to other reported results because in the related literature we did not find any estimates of MCID for kinematic parameters of lower limb joints post-stroke.

The strength of our study lies in the fact that four statistical methods were applied in determining MCID, significantly increasing the likelihood that the computed values are valid and reliable. By using the four methods and then choosing the highest estimated MCID value we could ensure that, ultimately, the MCID represents the patients’ perception of the change in knee sagittal ROM, it exceeds the measurement error, and reflects a significant correlation with the highly reliable BI [31], and accounts for the optimum trade-off between sensitivity and specificity. Notably, the differences in the estimates and the rationale for using the four methods are linked with the term “Clinically Important”, a part of the MCID acronym. “Clinically important” is not a strictly defined concept (there is no related formula). Each of these methods understands the term in a different way, and the related approaches, besides some advantages, present certain drawbacks. In the anchor-based method, this is the external criterion, which may be subjective. In the distribution-based method, this is the distribution of values in a sample. The sample may be small and not quite representative, and the results may be affected by unknown factors. In the linear regression method, this is the MCID of another tool, determined using one of these methods and presenting drawbacks of that method. In the ROC method, this is the “trade-off between sensitivity and specificity”. However, the trade-off may not be a good solution, e.g., because sensitivity is more important than specificity. Furthermore, the method also applies an external criterion from the anchor-based method, in a way “inheriting” its drawbacks [36,38,40].

Likewise, Hsieh et al. applied both anchor-based and distribution-based approaches to determine the MCID of BI post-stroke; they chose the larger MCID value of the two estimates. However, the study groups differed in terms of recovery stage and disability level [32]. Conversely, the current study applied all methods to one group of patients at the same recovery stage and disability level.

Notably, the anchor-based estimate of the MCID for knee flexion/extension ROM (8.48° affected/6.81° unaffected side) is higher than the distribution-based estimate (1.86° affected/5.63° unaffected side). This means that the value reflecting patients’ self-reported assessment of important change exceeds the measurement error, which shows that the MCID estimate is correct [34]. In other words, if the measurement error of a tool is higher than patients’ ratings, the tool cannot reliably assess the patients’ perceptions, making such an MCID questionable. Furthermore, the linear regression-based estimate of the MCID for knee flexion/extension ROM for affected side (7.71°) is similar to the value of the anchor-based estimate, while the one for unaffected side (4.66°) is closer to the distribution-based estimate. This suggests that the MCID of BI applied here is more specific to the affected than the unaffected side, which seems reasonable since BI measures the patient’s condition for which the affected side is mainly responsible. The identified cut-off points for the affected and unaffected side, i.e., 3.9° and 3.8°, respectively, are practically identical. This shows that the subjective rating of improvement applied here is impacted by a change in ROM on either side in a similar way. This may be surprising because a change on the affected side would seem more important for a subjective assessment of change. On the other hand, this observation may reflect compensatory mechanisms developing in gait patterns post-stroke [42].

The current study showed that the participants’ mean knee flexion/extension ROM for the affected and the unaffected side amounted to 34.62° and 42.33°, respectively. Similar values of knee sagittal ROM (33.7° and 49.8°, respectively) in individuals with chronic stroke were reported by Kim and Eng in a study focusing on 3D kinematic gait profiles [24]. Likewise, Carmo et al. in a 3D kinematic gait analysis of patients with chronic stroke showed similar values of knee sagittal ROM, 37.8° for the affected and 55.2 for unaffected side [43].

By assumption, the area under an ROC curve (AUC) is a measure of the overall usefulness of a test, with a greater area reflecting higher usefulness [38,41]. A larger AUC corresponds to greater effectiveness of a variable (here: knee ROM) in determining the improvement. An ideal ROC curve has an AUC of 1 [40]. In our study the AUC for affected side amounted to 0.883, a fairly good result. For the unaffected side it is only 0.64 which means it is far less suitable to identify improvement. This seems logical and is consistent with our earlier research on Gait Deviation Index (GDI), calculated from selected kinematic gait parameters, including sagittal knee ROM, in patients with chronic stroke. We showed that in these patients, gait analysis and the use of GDI could be limited to the affected leg or mean GDI. However, to identify any potential compensations, it is also necessary to calculate the GDI for unaffected leg [42].

Clinical relevance of the findings may be described by reference to intervention outcomes reported in the related literature. Drużbicki et al. assessed effects of treadmill training with/without biofeedback, including changes in kinematic gait parameters, in patients with chronic stroke. The therapy led to significant improvement in knee sagittal ROM on the affected and unaffected side by 4.6° (*p* = 0.0171) and 1.5° (*p* = 0.0007), respectively, in the biofeedback group, and by 3.4° (*p* = 0.0077) and 2.8° (*p* = 0.0152), respectively, in the non-biofeedback group [44]. However, by reference to the identified MCID one could conclude that the change was not clinically important. In a real-life situation, when changes in a patient’s condition are evaluated over longer timeframes, this would be a helpful conclusion, showing a need to continue a therapy or perhaps redesign the rehabilitation strategy adopted for a specific patient.

### Study Limitation

The shortcomings of the study include the fact that the MCIDs for knee flexion/extension ROM were determined only for patients at a chronic stage post-stroke. Generally, the largest progress in recovery of neuromotor functions is frequently observed during the early period post-stroke [45,46] due to the fact that recovery-related changes within the ischemic penumbra adjacent to the focal lesion occur relatively rapidly following onset, and later the process slows down [47,48]. Therefore, we suspect that the MCIDs determined for chronic stage may be lower than those relevant to acute phase post-stroke. This effect may also be associated with patients’ adaptation to and prolonged use of compensatory gait patterns. Hence, further research is needed to investigate this issue in patients at an acute phase after stroke.

Another limitation is that generalisation of the results may be farfetched due to the small size of the MCID group. It would be justified to conduct further research involving a larger sample of patients with varied motor control and presenting different gait patterns post-stroke. Furthermore, subsequent studies should aim to determine MCID for the kinematics of the hip, ankle and pelvis. It would also be worthwhile to determine the MCID for other kinematic parameters of the knee, for example peak knee flexion or knee angular velocity.

## 5. Conclusions

In summary, our findings provide the first estimates of MCID of the knee sagittal ROM for individuals with hemiparesis after stroke. We have estimated that, in patients with chronic stroke, MCID in the knee sagittal ROM for the affected and unaffected sides amount to 8.48° and 6.81°, respectively. From the viewpoint of clinical practice, the MCID facilitates assessment of effects produced by interventions administered to patients and may be used in designing treatment strategies. Therefore, these findings will assist clinicians and researchers in the interpretation of changes observed in kinematic sagittal plane parameters of the knee and their significance, and will be helpful in setting goals for patients with hemiparesis at a chronic stage post-stroke.

## Figures and Tables

**Figure 1 jcm-09-03305-f001:**
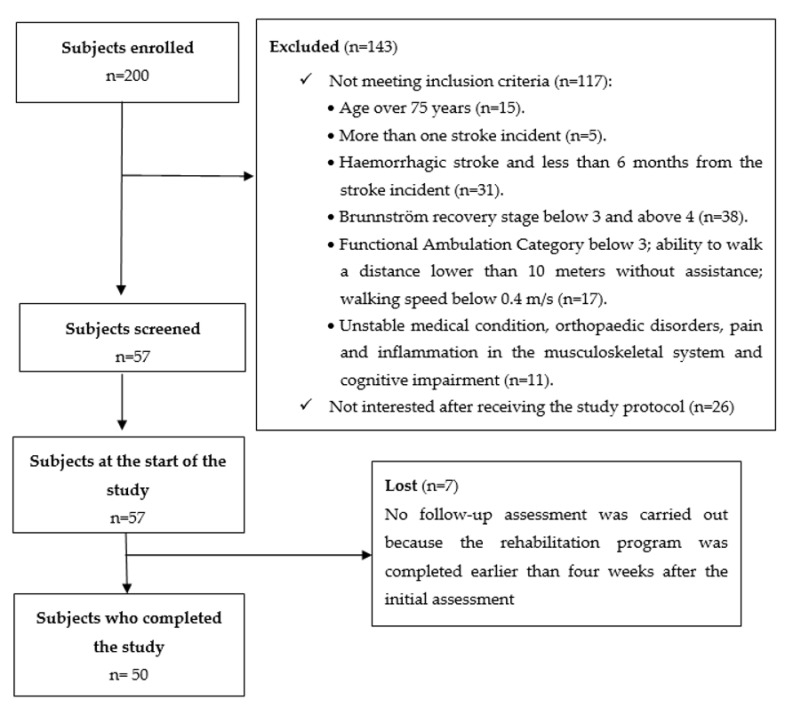
Flow of subjects through the study.

**Figure 2 jcm-09-03305-f002:**
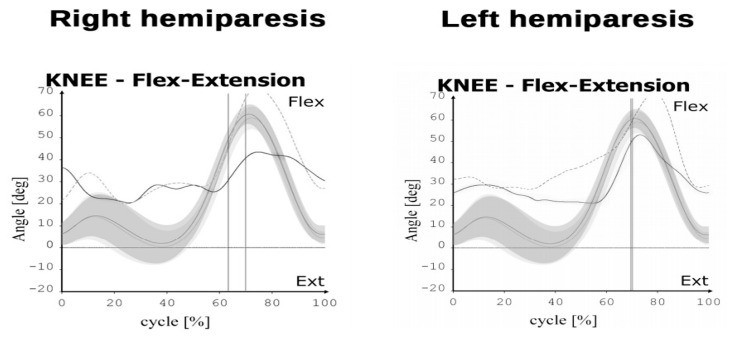
Representative figures for knee flexion/extension ROM for the affected and unaffected sides in two subjects with right- and left-hemiparesis. Average normal kinematics is represented by the grey solid line. The affected lower limb is shown by the black solid line. The unaffected lower limb is shown by the grey dotted line.

**Figure 3 jcm-09-03305-f003:**
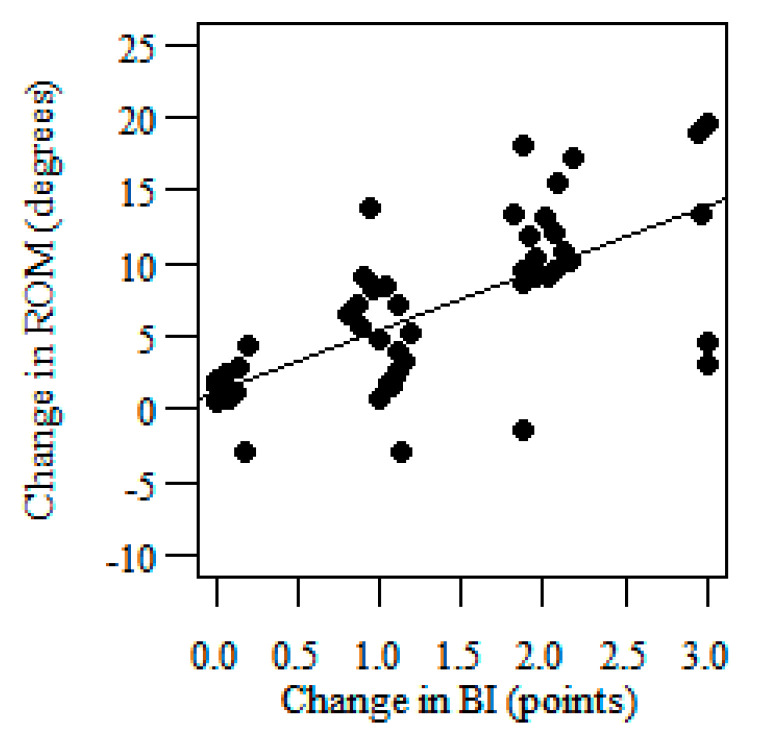
Scatterplots with linear regression showing the correlation between the change in the knee sagittal ROM (dependent variable) for the affected side and the change in the Barthel Index (BI) (independent variable).

**Figure 4 jcm-09-03305-f004:**
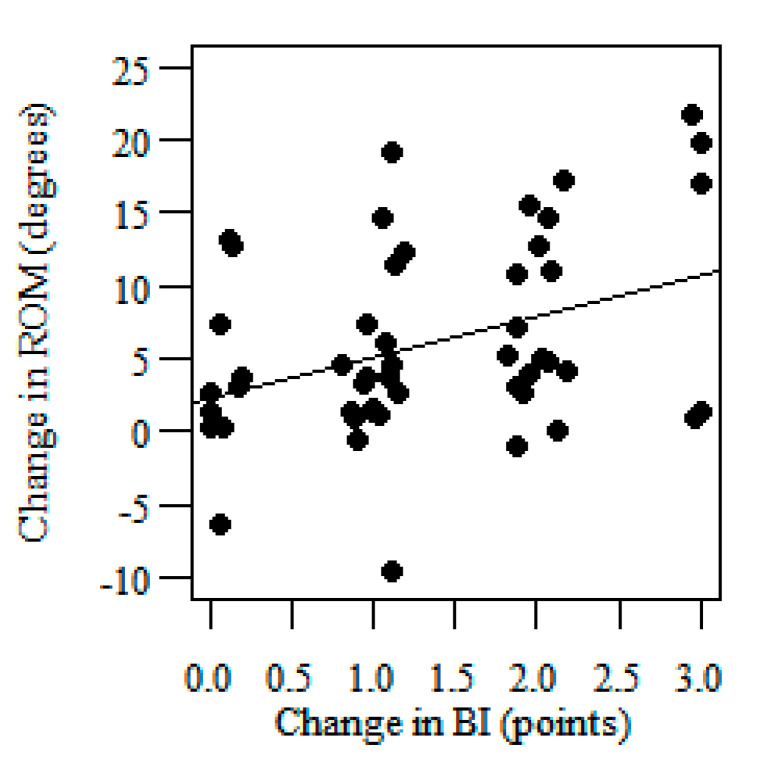
Scatterplots with linear regression showing the correlation between the change in the knee sagittal ROM (dependent variable) for the unaffected side and the change in the BI (independent variable).

**Figure 5 jcm-09-03305-f005:**
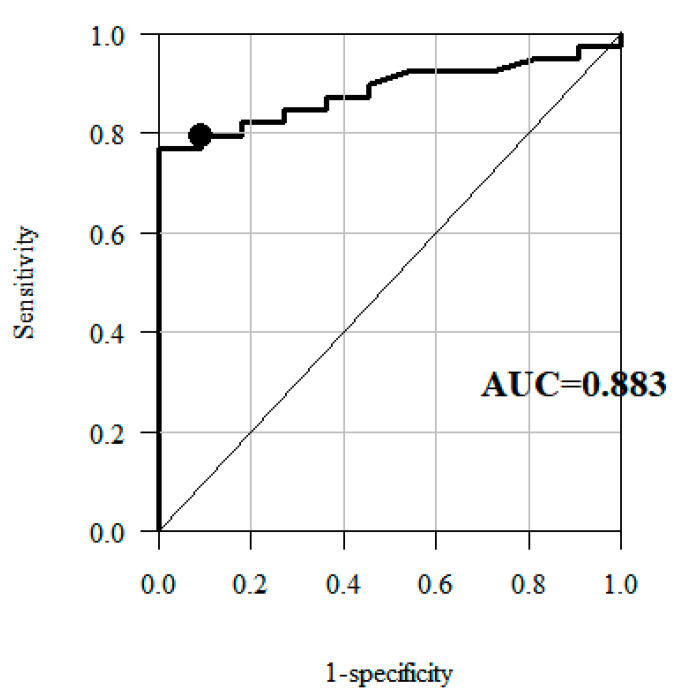
Receiver operating characteristic (ROC) curves—plot of sensitivity versus 1—specificity values for the knee sagittal ROM for the affected side, showing the trade-off possible between sensitivity and specificity, AUC—Area Under Curve (an ideal ROC curve has an AUC of 1), optimal cut-off point (black point in the plot).

**Figure 6 jcm-09-03305-f006:**
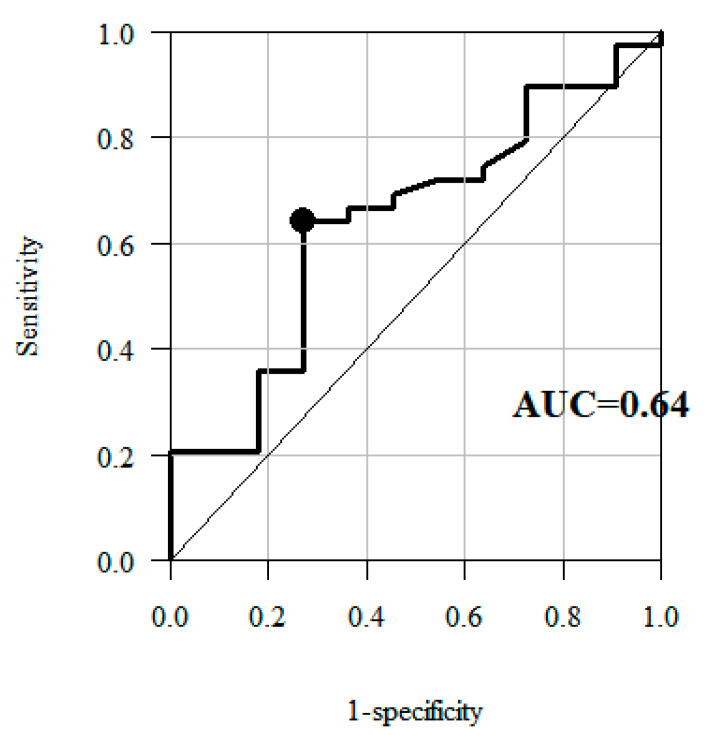
Receiver operating characteristic (ROC) curves—plot of sensitivity versus 1—specificity values for the knee sagittal ROM for the unaffected side, showing the trade-off possible between sensitivity and specificity, AUC—Area Under Curve (an ideal ROC curve has an AUC of 1), optimal cut-off point (black point in the plot).

**Table 1 jcm-09-03305-t001:** Baseline subject characteristics.

Group *N* = 50	
Mean age, (SD)	60.9 ± 11.2
Mean time from stroke (months), range	42 (8–120)
Sex (women/men)	18/32
Hemisphere lesions (right/left)	15/35

*N*—number of subjects, SD—standard deviation.

**Table 2 jcm-09-03305-t002:** Knee kinematic characteristics of the study participants.

Knee Flexion/Extension Range of Motion, *N* = 50	
Baseline ROM affected side (deg.), mean (SD)	34.62 (9.71)
Follow-up ROM affected side (deg.), mean (SD)	41.58 (9.52)
Baseline ROM unaffected side (deg.), mean (SD)	42.33 (8.52)
Follow-up ROM unaffected side (deg.), mean (SD)	48.78 (8.84)
Baseline ROM right side (deg.), mean (SD)	36.9 (10.24)
Follow-up ROM right side (deg.), mean (SD)	44.63 (10.2)
Baseline ROM left side (deg.), mean (SD)	40.08 (9.44)
Follow-up ROM left side (deg.), mean (SD)	45.43 (9.67)

*N*—number of subjects, SD—standard deviation, ROM—range of motion, deg.—degrees.

**Table 3 jcm-09-03305-t003:** The mean knee sagittal ROM for the affected/unaffected side in the minimal clinically important differences (MCID) group and no-change group.

**Mean Knee Sagittal ROM—MCID Group, *N* = 34**	
Baseline ROM affected side (deg.), mean (SD)	33.6 (9.34)
Follow-up ROM affected side (deg.), mean (SD)	42.1 (9.2)
Baseline ROM unaffected side (deg.), mean (SD)	41.64 (8.95)
Follow-up ROM unaffected side (deg.), mean (SD)	48.45 (9.25)
**Mean Knee Sagittal ROM—No-change Group, *N* = 16**	
Baseline ROM affected side (deg.), mean (SD)	38.23 (11.0)
Follow-up ROM affected side (deg.), mean (SD)	39.8 (11.26)
Baseline ROM unaffected side (deg.), mean (SD)	44.88 (7.77)
Follow-up ROM unaffected side (deg.), mean (SD)	48.6 (9.55)

*N*—number of subjects, SD—standard deviation, ROM—range of motion, deg.—degrees.

**Table 4 jcm-09-03305-t004:** MCID identified with four methods for knee flexion/extension ROM (affected side).

	MCID (deg.)	95% Confidence Interval	
Anchor-Based Method	8.48	6.7	10.26
Distribution-Based Method	1.86	1.3	3.27
Linear Regression Analysis	7.71	5.22	10.2
Receiver operating characteristic Curve	3.9	---	---

**Table 5 jcm-09-03305-t005:** MCID identified with four methods for knee flexion/extension ROM (unaffected side).

	MCID (deg.)	95% Confidence Interval	
Anchor-Based Method	6.81	4.57	9.05
Distribution-Based Method	5.63	3.93	9.88
Linear Regression Analysis	4.66	0.95	8.37
Receiver operating characteristic Curve	3.8	---	---

**Table 6 jcm-09-03305-t006:** Distribution of the knee flexion/extension ROM in the patients relative to age, in a dichotomous classification into age groups—prior to the rehabilitation versus rehabilitation effects.

**Knee Flexion/Extension ROM (before Rehabilitation)**	**Age Groups (Years)**						***p***
	**Up to 50**			**Over 50**			
	**Mean**	**95% CI**	**SD**	**Mean**	**95% CI**	**SD**	
Knee flexion/extension ROM affected side (deg.)	37.66	25.28–50.03	13.38	34.13	31.29–36.97	9.21	0.7353
Knee flexion/extension ROM unaffected side (deg.)	45.16	36.8–53.52	9.04	41.9	39.22–44.58	8.71	0.1762
**Knee Flexion/Extension ROM (Effects of Rehabilitation)**	**Age Groups (Years)**						***p***
	**Up to 50**			**Over 50**			
	**Mean**	**95% CI**	**SD**	**Mean**	**95% CI**	**SD**	
Knee flexion/extension ROM affected side (deg.)	4.56	0.15–9.26	5.09	7.35	5.58–9.12	5.75	0.2367
Knee flexion/extension ROM unaffected side (deg.)	5.01	3.9–13.12	8.76	6.32	4.33–8.30	6.44	0.3980

ROM—range of motion, deg.—degrees, CI—confidence interval, SD—standard deviation, *p*—test probability.

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
