# Peer review of "Estimating Minimal Clinically Important Differences for Knee Range of Motion after Stroke"

_jcm, 2020, doi:10.3390/jcm9103305_

Round 1

Reviewer 1 Report

The manuscript titled, “Estimating minimal clinically important differences 3 for knee range of motion after stroke “ by Guzik and colleagues provides minimal clinically relevant difference in for knee range of motion in the sagittal plane between affected and unaffected sides in post-stroke individuals. Multiple statistical methods have been employed to improve the accuracy of their findings. The reported clinical parameters for determination of improvement in kinematic parameters for knee, will be useful for clinicians when conducting their studies in post-stroke individuals. The paper is well-written and the method of statistical analysis is comprehensive, which further strengthens the manuscript. However, some concerns need to be addressed before publication.

  • Justification for the focus only on the sagittal knee ROM is not the strongest. For instance, in Lines 74-75, the systematic review that is referenced (16) in the manuscript clearly lists not just knee, but also pelvic, in the sagittal plane as strong indicators. Citation 20 also lists both knee and hip in the sagittal planes as having high reliability indices. If anything, multiple indices could have been employed.
  • ROM is abbreviated in line 106, but is listed much earlier in lines 71 and 72.
  • The paragraphs on advantages of the statistical methods employed in this manuscript is written in detail in the introduction section (lines 88-105). This seems misplaced. Please move it to the methods section, or results or even the discussion sections.
  • 200 patients were initially enrolled, but 143 were excluded. The flowchart lists that 117 did not meet the inclusion criteria. Then the sentence after that is incomplete. Please rectify the errors in the flowchart and list exactly what the inclusion criteria were, and why 143 were excluded when only 117 did not meet inclusion criteria. Also these numbers are mentioned in the flowchart, but are missing from the text. The section on participants starts off from the 50 that completed the study.
  • While the paper heavily focusses on statistical analysis, the data collection part is not described in much detail. A table on the subject characteristics would be required. Additionally, a more detailed account of how the kinematic protocol was designed would also be needed. Also the raw data obtained from the affected and the unaffected sides (representative figures), would be useful to the readers.

Author Response

Dear Reviewer,        

We thank you for reviewing our article titled, “Estimating minimal clinically important differences for knee range of motion after stroke”. We have made every effort to improve our manuscript, as guided by the reviewer’s helpful suggestions.

We thank the reviewer of all the comments. Answers are summarised below. All changes are highlighted as red text in the manuscript.

We hope you will be pleased with the changes, and support the publication of our revised manuscript.

With kind regards,

The authors of the article

Point 1: The manuscript titled, “Estimating minimal clinically important differences 3 for knee range of motion after stroke“ by Guzik and colleagues provides minimal clinically relevant difference in for knee range of motion in the sagittal plane between affected and unaffected sides in post-stroke individuals. Multiple statistical methods have been employed to improve the accuracy of their findings. The reported clinical parameters for determination of improvement in kinematic parameters for knee, will be useful for clinicians when conducting their studies in post-stroke individuals. The paper is well-written and the method of statistical analysis is comprehensive, which further strengthens the manuscript. However, some concerns need to be addressed before publication.

Justification for the focus only on the sagittal knee ROM is not the strongest. For instance, in Lines 74-75, the systematic review that is referenced (16) in the manuscript clearly lists not just knee, but also pelvic, in the sagittal plane as strong indicators. Citation 20 also lists both knee and hip in the sagittal planes as having high reliability indices. If anything, multiple indices could have been employed.

Response 1: Thank you for this valuable comment. We fully agree with the Reviewer’s opinion that, in addition to knee, the kinematics of both the hip and pelvis are strong predictors of walking performance post-stroke. However, MCID has not been determined yet for any of these. In view of the fact that the sagittal knee pattern in individuals with hemiparesis after stroke strongly correlates with the sagittal ankle or foot patterns and to a lesser degree to the sagittal hip pattern (Beyaert et al. 2015; Kaczmarczyk et al. 2009; Kinsella and Moran 2008; Mulroy et al. 2003; Olney et al. 1998), and given the importance of knee sagittal kinematic parameters, as a predictor of walking performance post-stroke, which was emphasised by numerous researchers (Olney et al. 1991; De Quervain et al. 1996; Kim and Eng 2004;  Mulroy et al. 2003; Kaczmarczyk et al. 2009; Boudarham et al. 2013), we decided to first focus on the MCID for knee flexion/extension range of motion. Nevertheless, we agree that further research should focus on the parameters listed by the Reviewer, and we mention that in the discussion (study limitations). In fact, this highly relevant Reviewer comment provides an excellent rationale for further research to determine the MCID for the ROM in other joints of the lower limbs. We believe this will be of great importance for the clinical practice.

BEFORE

lines 71-81: We focused only on the sagittal knee ROM for a few reasons; firstly, the sagittal knee pattern permanently affects post-stroke gait and is a predictor of walking performance post-stroke [18,20,23-25] Secondly, some of the most reliable results in terms of 3D kinematic parameters were found in the knee in the sagittal plane [16]. Thirdly, the importance of knee sagittal kinematic parameters, as a predictor of walking performance post-stroke, was emphasised by numerous researchers [18,20,24,25]. Fourthly, but most importantly, the sagittal knee pattern in individuals with hemiparesis after stroke strongly correlates with the sagittal ankle or foot patterns and to a lesser degree to the sagittal hip pattern [20,25,26]. However, a review of the related literature showed that no studies have focused on identifying the MCID for kinematic sagittal parameters of the knee post-stroke. This conclusion provided a motivation for the present study.

lines 330-334: Another limitation is that generalisation of the results may be farfetched due to the small size of the MCID group. It would be justified to conduct further research involving a larger sample of patients with varied motor control and presenting different gait patterns post-stroke. It would also be worthwhile to determine the MCID for other kinematic parameters of the knee, for example peak knee flexion or knee angular velocity.

AFTER

lines 71-81: It has been established that the highest reliability indices occur in the hip and knee in the sagittal plane [16,20], with the lowest errors in pelvic rotation and obliquity as well as hip abduction [16]. However, MCID has not yet been calculated for any of the above major predictors of walking performance post-stroke. Due to this we have decided to start the related research by identifying the MCID for knee flexion/extension range of motion (ROM) for the affected and the unaffected side. We chose to focus on this parameter first because it permanently affects post-stroke gait and numerous researchers have emphasised the importance of knee sagittal kinematic parameters, as a predictor of walking performance post-stroke [18,20,23-26]. Moreover, the sagittal knee pattern in individuals with hemiparesis after stroke strongly correlates with the sagittal ankle or foot patterns and to a lesser degree to the sagittal hip pattern [20,22,25,27,28]. This conclusion provided a motivation for the present study.

lines 403-408: Another limitation is that generalisation of the results may be farfetched due to the small size of the MCID group. It would be justified to conduct further research involving a larger sample of patients with varied motor control and presenting different gait patterns post-stroke. Furthermore, subsequent studies should aim to determine MCID for the kinematics of the hip, ankle and pelvis. It would also be worthwhile to determine the MCID for other kinematic parameters of the knee, for example peak knee flexion or knee angular velocity.

Point 2: ROM is abbreviated in line 106, but is listed much earlier in lines 71 and 72.

Response 2: Thank you for the helpful suggestion, the comment has been taken into account.

BEFORE

lines 71-72: This study was designed to identify the MCID for knee flexion/extension ROM for the affected and the unaffected side.

line 106: The study aimed to estimate MCID values for knee range of motion (ROM) in the sagittal plane

AFTER

lines 74-75: Due to this we have decided to start the related research by identifying the MCID for knee flexion/extension range of motion (ROM) for the affected and the unaffected side.

line 82: The study aimed to estimate MCID values for knee ROM in the sagittal plane

Point 3: The paragraphs on advantages of the statistical methods employed in this manuscript is written in detail in the introduction section (lines 88-105). This seems misplaced. Please move it to the methods section, or results or even the discussion sections.

Thank you for helpful suggestions. We have moved the paragraphs presenting the advantages of the statistical methods to methods section.

BEFORE

Introduction

lines 82-105: The MCID can be determined via anchor-based or distribution-based methods, linear regression analysis and specification of the receiver operating characteristic (ROC) curve. Anchor-based methods take into account change in scores related to a clearly defined clinical observation. External criteria applied include perception of the change by the patient or clinician. Besides a simple estimate of change, the construct of important change identifies that a change has occurred and is perceived as important by the patient, physician, or researcher [27-30].

Distribution-based methods take into account the statistical characteristics of the scores obtained, i.e. their significance, or sample variation, or measurement precision. Representing the latter type, i.e. the method of the standard error of measurement (SEM) is most promising for MCID-related research for three reasons. It takes into account both the amount of error specific to the instrument and the amount of random variation to be expected in repeated administrations. It is not greatly affected by the sample size or change variability. Finally, it is sample-independent [31,32].

Linear regression is applied in various analyses. The biggest advantage of linear regression models is linearity: it makes the estimation procedure simple and, most importantly, these linear equations are easy to interpret on a modular level [33,34].

ROC curves are often applied to visualise the connection/trade-off between sensitivity and specificity for every possible cut-off for a test or a combination of tests. Sensitivity is described as a probability that if a rule says an event will occur, it indeed will occur. Specificity on the other hand is a probability that if a rule says an event will not happen, it indeed will not happen. When we have calculated sensitivity and specificity, we may draw ROC curve for every possible cut-off. Generally, it is impossible to have high sensitivity and high specificity at the same time, but we strive for perfection. Hence, we want our ROC curve to get as close as possible to the left upper corner, to identify a point in that area. In terms of sensitivity and specificity, this point most accurately corresponds to the change identified as MCID [35,36].

AFTER

Methods

2.3. Data analysis  

lines 189-227: The MCID for the knee ROM, for the affected/unaffected side, was determined using four methods, and finally the highest result was selected.

Anchor-based method made it possible to identify the first estimate for the MCID, i.e. the mean change in the knee sagittal ROM for the affected/unaffected side in the “positive change group” (MCID group). Anchor-based methods take into account change in scores related to a clearly defined clinical observation. External criteria applied include perception of the change by the patient or clinician. Besides a simple estimate of change, the construct of ‘important change’ implies that a change has occurred and is perceived as significant by the patient, physician, or researcher [33-36].

Distribution-based method was used to determine the second estimate for the MCID, i.e. the SEM was computed as the square root of the variance of a change in the knee sagittal ROM in the relevant subgroup. Distribution-based methods take into account the statistical characteristics of the scores obtained, i.e. their significance, or sample variation, or measurement precision. Representing the latter type, the method of the standard error of measurement (SEM) is most promising for MCID-related research for three reasons. It takes into account both the amount of error specific to the instrument and the amount of random variation to be expected in repeated administrations. It is not greatly affected by the sample size or change variability. Finally, it is sample-independent [32,37].

Linear regression analysis was applied to identify the third estimate for the MCID, i.e. the relationship between the change in the knee sagittal ROM (dependent variable) and the change in BI (independent variable). Linear regression is applied in various analyses. The biggest advantage of linear regression models is linearity: it makes the estimation procedure simple and, most importantly, these linear equations are easy to interpret on a modular level [38,39].

ROC curve, applied to determine the fourth estimate of the MCID in the knee ROM for the affected/unaffected side, made it possible to identify the optimal cut-off point for the change in the knee sagittal ROM, producing the optimum relation of sensitivity and specificity. ROC curves are often applied to visualise the connection/trade-off between sensitivity and specificity for every possible cut-off for a test or a combination of tests. Sensitivity is described as a probability that if a rule says an event will occur, it indeed will occur. Specificity on the other hand is a probability that if a rule says an event will not happen, it indeed will not happen. When we have calculated sensitivity and specificity, we may draw ROC curve for every possible cut-off. Generally, it is impossible to have high sensitivity and high specificity at the same time, but we strive for perfection. Hence, we want our ROC curve to get as close as possible to the left upper corner, to identify a point in that area. In terms of sensitivity and specificity, this point most accurately corresponds to the change identified as MCID [40,41].

Point 4: 200 patients were initially enrolled, but 143 were excluded. The flowchart lists that 117 did not meet the inclusion criteria. Then the sentence after that is incomplete. Please rectify the errors in the flowchart and list exactly what the inclusion criteria were, and why 143 were excluded when only 117 did not meet inclusion criteria. Also these numbers are mentioned in the flowchart, but are missing from the text. The section on participants starts off from the 50 that completed the study.

Response 1: Thank you for this valuable comment. We have rectified the errors in the flowchart and listed the detailed inclusion criteria. We have also added detailed description of the flow of the subjects through the study with the relevant numbers, in the text.

BEFORE

2.1. Participants

lines 112-115: Fifty patients with stroke in a chronic phase of recovery (mean age 60.9 ± 11.2 years; mean time from stroke 42 months, ranging from 8 to 120 months; 18 females, 32 males; 15 patients with right hemisphere lesions, 35 patients with left hemisphere lesions), were identified in a database of a rehabilitation clinic

AFTER

2.1. Participants

lines 89-94: The study involved 57 patients in a chronic phase of recovery post stroke recruited among the 200 patients with stroke receiving treatment at a rehabilitation clinic (117 patients were not eligible to participate and 26 patients were not interested after receiving information about the study protocol). Ultimately the analyses took into account 50 individuals (7 patients were not assessed at the follow-up because their rehabilitation program was completed earlier than four weeks after the initial assessment).

Point 5: While the paper heavily focusses on statistical analysis, the data collection part is not described in much detail. A table on the subject characteristics would be required. Additionally, a more detailed account of how the kinematic protocol was designed would also be needed. Also the raw data obtained from the affected and the unaffected sides (representative figures), would be useful to the readers.

Response 1: Thank you for this helpful suggestion. We have added description of the data collection process and the way the kinematic protocol was designed. We have also added the table with the subject characteristics and the representative figures.

BEFORE

2.2. Measures

lines 130-148: Kinematic knee data were collected with a six-camera motion capture system (BTS SMART-DX 700, 250 Hz) with software, in SMART Capture, Tracker and Analyzer and two force-plates. Passive markers were placed on the subjects’ skin, in accordance with the internal protocol of the system Davis Marker Placement on the sacrum, pelvis, femur, fibula, foot [27]. Each 3D assessment was preceded by the system calibration. The recording of each patient covered a walking distance of 10 m, repeated at least six times. More trials were necessary if the patient lost balance or excessively hesitated during the basic trials.  The tests were conducted without shoes. The subjects were asked to walk the distance at their natural pace. A 3D skeletal model was created for each subject. The model and the joint centres were scaled, taking into account the subject’s height and weight. The generic model also was scaled. After the tests were performed, the data were collected and processed with software from the BTS Smart system (Smart Tracker and Smart Analyzer). The analyses took into account the complete range of knee flexion and extension in a gait cycle for the affected and the unaffected side.

AFTER

2.2. Measures

lines 159-178: Kinematic knee data were collected with a six-camera motion capture system (BTS SMART-DX 700, 250 Hz) with software, in SMART Capture, Tracker and Analyzer and two force-plates. Passive markers were placed on the subjects’ skin, following the internal protocol of the system Davis Marker Placement, on the sacrum, pelvis (the anterior and posterior iliac spine), femur (lateral epicondyle, great trochanter and in the lower one-third of the shank), fibula (lateral malleolus, lateral end of the condyle in the lower one-third of the shank), as well as foot (metatarsal head and heel) [29]. Each 3D assessment was preceded by the system calibration. The recording of each patient covered a walking distance of 10 m, repeated at least six times. More trials were necessary if the patient lost balance or excessively hesitated during the basic trials. The tests were conducted without shoes. The subjects were asked to walk the distance at their natural pace. A 3D skeletal model was created for each subject. The model and the joint centres were scaled, taking into account the subject’s height and weight. The generic model also was scaled. After the tests were performed, the data were collected and processed with software from the BTS Smart system (Smart Tracker and Smart Analyzer). The analyses took into account the complete range of knee flexion and extension in a gait cycle for the affected and the unaffected side. The gait cycle for each leg was defined to comprise all the phases starting with heel strike and ending with the next contact of the same foot with the ground. One stance phase and one swing phase were recorded during a single gait cycle performed by each leg. The analyses took into account a minimum of six gait cycles performed by each subject. Based on that, mean values of biomechanical gait parameters were calculated for the complete range of knee flexion and extension for the affected and unaffected side.

2.1. Participants

lines 106: The baseline subject characteristics are shown in Table 1.

Table 1. Baseline subject characteristics.

Group N=50

Mean age, (SD)

60.9 ± 11.2

Mean time from stroke (months), range

42 (8-120)

Sex (women/men)

18/32

Hemisphere lesions (right/left)

15/35

N – number of subjects, SD – standard deviation

lines 107-109: Representative graphs for knee flexion/extension ROM for the affected and unaffected sides in two subjects with right- and left-hemiparesis are shown in Figure 2.

Reviewer 2 Report

Review for Manuscript No. JCM-949015 entitled “Estimating minimal clinically important differences for knee range of motion after stroke”

The manuscript provides estimation on the minimal clinically important differences (MCID) for knee sagittal range of motion (ROM) for the affected and unaffected sides in fifty individuals at chronic stage post-stroke. The authors used methods including anchor-base method, distribution-base method, linear regression analysis, and specification of the receiver operating characteristic curve to estimate MCID values. The research method is new and reliable, and the result are valuable and attractive to researchers and clinicians because it can be used to interpret changes observed in kinetic sagittal ROM as well as to set goals for post-stroke patients. The manuscript was well written,  One small suggestion to correct on the typo: Page 4, Figure 4, second box from the top, the sentence/phase “not interested after receiving the” is incomplete.  

Author Response

Dear Reviewer,        

We thank you for reviewing our article titled “Estimating minimal clinically important differences for knee range of motion after stroke”.

We thank the reviewer of the comment. Answers are summarised below. The changes are highlighted as red text in the manuscript.

We hope you will be pleased with the changes, and support the publication of our revised manuscript.

With kind regards,

The authors of the article

Point 1: The manuscript provides estimation on the minimal clinically important differences (MCID) for knee sagittal range of motion (ROM) for the affected and unaffected sides in fifty individuals at chronic stage post-stroke. The authors used methods including anchor-base method, distribution-base method, linear regression analysis, and specification of the receiver operating characteristic curve to estimate MCID values. The research method is new and reliable, and the result are valuable and attractive to researchers and clinicians because it can be used to interpret changes observed in kinetic sagittal ROM as well as to set goals for post-stroke patients. The manuscript was well written, One small suggestion to correct on the typo: Page 4, Figure 4, second box from the top, the sentence/phase “not interested after receiving the” is incomplete.  

Response 1: Thank you for this valuable comment. In accordance with the Reviewer suggestion we have corrected the typo.

Reviewer 3 Report

In the present paper Guzik et al.  suggest that in chronic stroke, MCID estimates of knee sagittal ROM for the affected side amount to 8.48° and for
the unaffected side to 6.81°. The authors claimj that these findings could assist clinicians and researchers in interpreting the significance of changes observed in kinematic sagittal plane parameters of the knee.

The study is interesting and the conclusions are consistent with the described experiments.

My main concern is related to possible confounder factrs. In particular, the use of antinflammatory drugs should be reported.

In addition a more appropriate age stratification is necessary. The renge 35-75 is too wide to make general conclusions.

Author Response

Dear Reviewer,        

We thank you for reviewing our article titled “Estimating minimal clinically important differences for knee range of motion after stroke”. We have made every effort to improve our manuscript, as guided by the reviewer’s helpful suggestions.

We thank the reviewer of all the comments. Answers are summarised below. All changes are highlighted as red text in the manuscript.

We hope you will be pleased with the changes, and support the publication of our revised manuscript.

With kind regards,

The authors of the article

Point 1: In the present paper Guzik et al.  suggest that in chronic stroke, MCID estimates of knee sagittal ROM for the affected side amount to 8.48° and for the unaffected side to 6.81°. The authors claimj that these findings could assist clinicians and researchers in interpreting the significance of changes observed in kinematic sagittal plane parameters of the knee.

The study is interesting and the conclusions are consistent with the described experiments.

My main concern is related to possible confounder factors. In particular, the use of antinflammatory drugs should be reported.

Response 1: Thank you, this is a good point. We fully agree with the Reviewer that the use of antinflammatory drugs may affect the gait pattern. Due to this we did not recruit any patients with orthopedic disorders, including individuals at the time experiencing pain in the musculoskeletal system and potentially needing to use antinflammatory drugs. We have revised the relevant section and added more detailed information related to the exclusion criteria, which besides orthopedic disorders refer to pain and inflammation in the musculoskeletal system significantly affecting gait and requiring anti-inflammatory drugs.

BEFORE

lines 119-121: Patients excluded from the study had more than one stroke incident, presented unstable medical condition, orthopaedic disorders of the lower limbs and cognitive impairment affecting their ability to understand the instructions, and perform the tasks.

AFTER

lines 99-101: Patients excluded from the study had more than one stroke incident, presented unstable medical condition, orthopaedic disorders of the lower limbs, pain and inflammation in the musculoskeletal system significantly affecting gait and requiring anti-inflammatory drugs, cognitive impairment affecting their ability to understand the instructions, and perform the tasks.

Point 2: In addition a more appropriate age stratification is necessary. The range 35-75 is too wide to make general conclusions.

Response 2: Thank you for this valuable comment. Given the fact that the subjects represented a wide age range, in order to acquire more reliable result we performed additional comparative analysis of the groups distinguished based on age stratification (dichotomous classification into two groups of subjects up to 50 and over 50 years of age). For this purpose, descriptive statistics were compared and differences between the average values were assessed (Mann-Whitney test) by examining the relations between age and the measures of knee flexion/extension ROM identified before the start of the rehabilitation program and showing effects of rehabilitation. The relevant findings were described in the Results section and Table 6 was added. The age groups did not differ before the rehabilitation and after it was completed (effect of rehabilitation), as a result it was possible to draw a general conclusion for the entire population.

BEFORE

-

AFTER

2.3. Data analysis

lines 222-227: In order to assess the age-related effects, comparative analyses were performed taking into account groups distinguished based on the category of age (dichotomous age stratification into two groups of subjects up to 50 and over 50 years of age). For this purpose, descriptive statistics were compared and differences between the average values were assessed (Mann-Whitney test) by examining the relations between age and the knee flexion/extension ROM identified before the start of the rehabilitation program and showing effects of rehabilitation.

  1. Results

lines 302-307: Given the fact that the subjects participating in the study represented a wide age range, they were divided into two groups – up to 50 (7 subjects -14%) and over 50 years of age (43 subjects – 86%). A comparison based on the dichotomous distinction of the age groups showed there were no relationships between age and the knee flexion/extension ROM in the patients prior to the rehabilitation (Table 6). Likewise, a comparative analysis of the results achieved by these two age groups showed no statistically significant differences in the effects of the rehabilitation (Table 6).

Table 6. Distribution of the knee flexion/extension ROM in the patients relative to age, in a dichotomous classification into age groups – prior to the rehabilitation versus rehabilitation effects.

Knee flexion/extension ROM
(before rehabilitation)

Age groups [years]

p

up to 50

over 50

Mean

95% CI

SD

Mean

95% CI

SD

Knee flexion/extension ROM affected side (deg.)

37.66

25.28-50.03

13.38

34.13

31.29-36.97

9.21

0.7353

Knee flexion/extension ROM unaffected side (deg.)

45.16

36.8-53.52

9.04

41.9

39.22-44.58

8.71

0.1762

Knee flexion/extension ROM

(effects of rehabilitation)

Age groups [years]

p

up to 50

over 50

Mean

95% CI

SD

Mean

95% CI

SD

Knee flexion/extension ROM affected side (deg.)

4.56

0.15-9.26

5.09

7.35

5.58-9.12

5.75

0.2367

Knee flexion/extension ROM unaffected side (deg.)

5.01

3.9-13.12

8.76

6.32

4.33-8.30

6.44

0.3980

ROM – range of motion, deg. – degrees, CI - confidence interval, SD – standard deviation, p – test probability